# F_4_-Neuroprostane Effects on Human Sperm

**DOI:** 10.3390/ijms24020935

**Published:** 2023-01-04

**Authors:** Elena Moretti, Cinzia Signorini, Daria Noto, Roberta Corsaro, Lucia Micheli, Thierry Durand, Camille Oger, Jean Marie Galano, Giulia Collodel

**Affiliations:** 1Department of Molecular and Developmental Medicine, Policlinico Le Scotte, University of Siena, 53100 Siena, Italy; 2Department of Medicine, Surgery and Neurosciences, University of Siena, 53100 Siena, Italy; 3Institut des Biomolécules Max Mousseron (IBMM), Pole Chimie Balard Recherche, UMR 5247, CNRS, Université de Montpellier, ENSCM, 34090 Montpellier, France

**Keywords:** acrosome reaction, capacitation, F_4_-neuroprostanes, human sperm, in vitro study, sperm motility, ryanodine receptor

## Abstract

Swim-up selected human sperm were incubated with 7 ng F_4_-neuroprostanes (F_4_-NeuroPs) for 2 and 4 h. Sperm motility and membrane mitochondrial potential (MMP) were evaluated. The percentage of reacted acrosome was assessed by pisum sativum agglutinin (PSA). Chromatin integrity was detected using the acridine orange (AO) assay and localization of the ryanodine receptor was performed by immunofluorescence analysis. Sperm progressive motility (*p* = 0.02) and the percentage of sperm showing a strong MMP signal (*p* = 0.012) significantly increased after 2 h F_4_-NeuroP incubation compared to control samples. The AO assay did not show differences in the percentage of sperm with dsDNA between treated or control samples. Meanwhile, a significantly higher number of sperm with reacted acrosomes was highlighted by PSA localization after 4 h F_4_-NeuroP incubation. Finally, using an anti-ryanodine antibody, the immunofluorescence signal was differentially distributed at 2 and 4 h: a strong signal was evident in the midpiece and postacrosomal sheath (70% of sperm) at 2 h, whereas a dotted one appeared at 4 h (53% of sperm). A defined concentration of F_4_-NeuroPs in seminal fluid may induce sperm capacitation via channel ions present in sperm cells, representing an aid during in vitro sperm preparation that may increase the positive outcome of assisted fertilization.

## 1. Introduction

Isoprostanoids are non-enzymatic oxygenated metabolites derived from polyunsaturated fatty acids (PUFA). They are named based on the compounds from which they are derived: arachidonic acid gives F_2_-isoprostanes (F_2_-IsoPs), adrenic acid gives F_2_-dihomo-isoprostanes (F_2_-dihomo-IsoPs), F_3_-isoprostanes (F_3_-IsoPs) are derived from eicosapentaenoic acid, and F_4_-neuroprostanes (F_4_-NeuroPs) are derived from docosahexaenoic acid [1,2].

F_4_-NeuroPs represent an index of oxidative stress in different pathological neurological diseases [3,4,5], as well as in intracranial aneurysm development [6]. Moreover, they have been suggested as a predictor of atherosclerosis prevention and as a mediator of cardioprotective antiarrhythmic properties, thus providing a marker of ischemic stroke [7,8,9,10].

NeuroPs have been reported in organic fluids such as plasma [11], urine [12], and, recently, semen [13] where they may represent an indicator of semen quality. Fertile men had lower levels of seminal F_4_-NeuroPs than infertile men with different pathological reproductive conditions and a positive correlation was observed with sperm necrosis [13]. The in vitro effects of F_4_-NeuroPs on sperm have been previously investigated; human sperm was incubated in vitro with different concentrations of chemically synthetized F_4_-NeuroPs (4-F_4t_-NeuroP and 10-F_4t_-NeuroP). Three F_4_-NeuroP concentrations were tested, and the reaction to a total quantity of 7 ng was progressive sperm motility enhancement, suggesting a possible role of F_4_-NeuroPs in biological processes and in the promotion of the capacitation process [14]. 

To obtain fertilization, sperm need to be capacitated, which involves sperm hyperactivation and acrosomal reaction [15].

Various characteristics of capacitated spermatozoa are influenced by the fluxes of different ions. The inducers of the acrosome reaction are coupled to complex calcium signalling. 

Pharmacological treatments have demonstrated that capacitation is associated with ryanodine receptors, CatSper, and store-operated calcium channels [16]. 

The aim of this paper was to investigate the biological role of F_4_-NeuroPs on spermatozoa capacitation in vitro.

The quantity of 7 ng F_4_-NeuroPs was incubated 2 and 4 h with swim-up selected human sperm, and motility parameters as well as membrane mitochondrial potential (MMP) were evaluated. The percentage of reacted acrosome was assessed by Pisum sativum agglutinin (PSA). Chromatin integrity was detected using the acridine orange assay. As a last step, we performed immunofluorescence localization of the ryanodine receptor. 

## 2. Results

Semen parameters were evaluated in ejaculated human spermatozoa from seven donors, including sperm concentration, motility, morphology, and vitality value of each sample resulted higher than 25th percentile [17].

The samples were subjected to swim-up, and the selected upper fraction was incubated with 7 ng F_4_-NeuroPs (1:1 mixture of 4-F_4t_-NeuroP and 10-F_4t_-NeuroP) for 2 and 4 h. Controls were incubated without F_4_-NeuroPs. Data on motility, obtained using a CASA, confirmed that sperm progressive motility significantly increased after 2 h F_4_-NeuroP (the 1:1 mixture above reported) incubation (*p* = 0.02, Table 1). Moreover, the CASA evaluation detected a displacement in the sperm head movement of the hyperactivated sperm, although the amplitude of lateral head displacement was not significant compared to that observed in control samples. Curvilinear velocity, straight-line velocity, average path velocity, linearity, and beat cross frequency showed no significant differences between treated and control samples.

After 4 h F_4_-NeuroP incubation, the sperm motility did not differ compared to that of the 4 h control (Table 1). 

In all of the experiments, sperm vitality (Eosin Y assay) was evaluated. No significant difference was found among the treated and control samples.

### 2.1. DNA Integrity Assessment with Acridine Orange (AO) Staining

The AO assay enables the analysis of DNA susceptibility to damage. Data were obtained as the percentage of sperm with double-stranded DNA (dsDNA, green fluorescence, normal) and single-stranded DNA (ssDNA, red fluorescence). The percentage of sperm with dsDNA in the swim-up fractions incubated for 2 and 4 h with F_4_-NeuroPs did not differ from that in the control samples (both up 98%). 

### 2.2. Mitochondrial Membrane Potential (MMP) Assessment with JC-1 Labeling

JC-1 labeling successfully differentiates between sperm with low (green fluorescence) and high (red fluorescence) mitochondrial membrane potential (MMP). This biofunctional parameter is strictly related to sperm motility, and the measurement of MMP accounts for the function of this intracellular organelle. We also evaluated the presence of strong JC-1 labeling (strong red signal).

After 2 h F_4_-NeuroP incubation, the percentage of sperm showing a red spot significantly increased compared to that in control samples (*p* = 0.012). In addition, after incubation for 2 h with F_4_-NeuroPs, the percentage of sperm displaying strong red signals appeared to be significantly higher than that in control samples (*p* = 0.002). After 4 h F_4_-NeuroP incubation, MMP in the treated group did not differ significantly from that in the control group. Lastly, the percentage of strong red signals was significantly increased in the 4 h F_4_-NeuroP group with respect to that in the 4 h control group (Table 2).

### 2.3. Pisum Sativum Agglutinin (PSA) Evaluation

Acrosome integrity was detected in 100 spermatozoa. Fluorescent acrosome labeling defined three patterns, including intact (in the apical portion of the sperm head, Figure 1a), reduced (acrosome partially labeled, Figure 1b,c), and absent (completely reacted, Figure 1b,c). 

The number of sperm with totally reacted acrosome increased with time. The acrosomes appeared intact at short times, as in the control samples or in the 2 h F_4_-NeuroP incubation samples. After 4 h F_4_-NeuroP incubation, the sperm acrosomes underwent exocytosis or were predisposed to this condition as the incubation continued. The data are shown in Table 3. 

### 2.4. Ryanodine Receptor Localization

Assuming that the effect of F_4_-NeuroP incubation may be mediated by the ryanodine receptor, we investigated its presence and localization using immunofluorescence analysis. The investigation involved sperm from control samples and after 2 and 4 h F_4_-NeuroP incubation.

The signal was differentially distributed. It was absent, diffuse, or fair, located in the mitochondrial sheath, postacrosomal region, or in the entire tail. 

In control samples, the sperm showed diffuse fair labeling (60%, Table 4, Figure 2a) and some sperm showed a spot in the centriolar region (25%, Figure 2b). A very small percentage of sperm showed mitochondrial localization (5%). When selected sperm were incubated for 2 h with F_4_-NeuroPs, a strong signal was evident in the midpiece and postacrosomal sheath (70%, Figure 2 c,d). It was possible to observe a reduced number of sperm with diffuse signals (10%) and/or dotted labeling (9%) in the entire tail. It is important to emphasize that variability between patient samples was observed, which was not particularly relevant since the experiment was carried out after swim-up. 

After 4 h F_4_-NeuroP incubation, dotted labeling was detected in 53% of sperm (Figure 2e,f)

## 3. Discussion

F_4_-NeuroPs may be involved in the induction of physiological activities. Previously, Signorini et al. [14] reported that a moderate level of F_4_-NeuroPs stimulated sperm motility hyperactivation in vitro, suggesting its role as an inducer of the capacitation process. 

In this paper, a possible mechanism of action of F_4_-NeuroPs on human sperm is suggested.

After swim-up selection, the sperm upper fraction was incubated with a defined concentration of F_4_-NeuroPs (7 ng as the final quantity) for 2 and 4 h. An increase in progressive sperm motility compared to control samples was displayed only by the 2 h incubation group. Sperm showed hyperactivation, supported by high MMP, suggesting that F_4_-NeuroPs could influence the physiological behavior of sperm. Mitochondrial functionality evaluated through MMP assessment has been associated with sperm quality. MMP was positively correlated with sperm motility and negatively correlated with lipid peroxidation in bovines [18]. In humans, MMP can be considered as a potential regulator and indicator of sperm motility and hence is directly related to male fertility [19]. Recently, a study simultaneously examined MMP and the acrosin activity of spermatozoa in a large population, observing that increased acrosin activity was closely associated with high MMP [20]. 

It is important to underline that the analysis of susceptibility to DNA degradation using the AO assay excluded any damage after F_4_-NeuroP incubation both at 2 and 4 h.

The biological influence of these molecules has been demonstrated in the literature. 4-F_4t_-NeuroP was able to protect ischemic/reperfusion cardiac injuries by regulating mitochondrial homeostasis [9] and showed anti-inflammatory properties [21]. 4-F_4t_-NeuroP was injected into neuroblastoma cells (SH-SY5Y) and rodents, demonstrating a neuroprotective effect by regulating the transcriptional level of the antioxidant enzyme heme oxygenase-1 [22]. Recently, 4(RS)-4-F_4t_-NeuroP reduced the levels of reactive oxygen species induced by lipopolysaccaharide (LPS) in primary and immortalized mouse microglia cells and decreased NFκB-p65 and iNOS and TNFα levels. Moreover, 4(RS)-4-F_4t_-NeuroP positively influenced the mitochondrial condition and upregulated the Nrf2/HO-1 antioxidative pathway [3].

The influence of inflammatory lipid mediator isoprostanes were also studied in human and murine urinary bladders ex vivo by Molnar et al. [23]. The results suggested that isoprostanes caused contraction, an effect mediated by thromboxane prostanoid receptors. A therapeutic treatment may involve the G12/13-Rho-Rho kinase signaling pathway in mediating the contraction.

In semen, the action of F_4_-NeuroPs was hypothesized to influence sperm capacitation [14] by acting as a specific sperm receptor. Spermatozoa capacitation predicts an ordered sequence of events in order to acquire fertilizing capacity. Sperm membrane components are reorganized and tyrosine phosphorylation occurs, which allows the acrosome reaction that represents the last step of this process [24]. The acrosome reaction is characterized by the exocytosis of the acrosomal content and the release of vesicles composed by part of outer acrosomal membrane and plasma membrane [25].

Our data suggested that a 2 h F_4_-NeuroP incubation stimulated a sperm hypermotility that was not present in controls. The acrosomes appeared intact in almost the totality of sperm. Then, at 4 h F_4_-NeuroP incubation, a peculiar increase in the percentage of reacted acrosomes was observed using PSA. PSA can be used to assess the acrosome status and identify damaged or reacted acrosomes, but in this paper, we evaluated swim-up selected sperm and the increased number of sperm without PSA labeling may indicate the induction of capacitation. 

In a previous paper, Signorini et al. [14] detected acrosomal localization of phospho-AMPKα after 4 h F_4_-NeuroP incubation. It is known that an increase in the phosphorylation of PKA substrates occurs during capacitation [26]. AMPK is a protein that regulates energy balance and metabolism, and it was recently identified in boar spermatozoa where it regulates key functional sperm processes essential for fertilization [27]. 

The timing of capacitation differs among men but is consistent [28]; consequently, in vitro interventions in the timing and mechanisms of capacitation could be an important goal during assisted fertilization procedures.

Moreover, capacitation is characterized by an increase in intracellular pH and calcium levels [29]. 

To hypothesize the mechanism of action of F_4_-NeuroPs, we reviewed the role of cardiac ryanodine receptor (RYR2), a critical component of Ca2+ handling machinery, which is responsible for β-adrenergic enhancement of cardiac contractility [30]. 

In this paper, the localization of RyR in sperm incubated with F_4_-NeuroPs for 2 and 4 h was investigated and the labeling appeared to be differentially distributed. A peculiar mitochondrial localization of RyR was highlighted after 2 h incubation. The mitochondrial involvement agreed with the MMP localization detected for the same incubation time. At 4 h incubation, the anti-ryanodine signal was identified along the entire tail and a reduction in acrosome integrity, evaluated with PSA, was observed. Previous in vitro experiments demonstrated, by MMP, that the functional integrity of mitochondria is required for acrosin activity [20]. 

Zhou et al. [31] reported that tripeptidyl peptidase II (TPPII) antagonists determined the increase in sperm intracellular Ca ^(2+)^ levels that positively influenced sperm characteristics. This TPPII effect may be counterbalanced by the inhibitors of RyRs, which are the main intracellular Ca ^(2+)^ channels. Using immunofluorescence, RyR3 was localized in the acrosomal region of mature sperm, suggesting that TPPII may influence sperm maturation by this receptor and its function.

Recently, sperm maturation and fertility have been coupled with the fatty acid composition of the sperm membrane [32]. Among fatty acids, DHA quantity was positively associated with sperm parameters [33]. Lower DHA concentrations were reported in asthenozoospermic and oligozoospermic men than in fertile men [34]. Dietary supplementation with DHA reduced sperm DNA fragmentation and increased seminal antioxidants potentiality [35]. Levels of oxidized DHA are associated with DHA levels and the clinical severity of some pathologies [7]; this could indicate a similar behavior in sperm cells. High DHA content in sperm not only guarantees normal sperm function but also the formation of its metabolite, F_4_-NeuroP, which plays a role in sperm capacitation.

## 4. Conclusions

In conclusion, a defined concentration of F_4_-NeuroPs in seminal fluid may induce sperm capacitation via channel ions present in sperm cells. Considering that infertile men have difficulties with capacitation, these molecules could, for some patients, aid in sperm preparation in vitro and increase the positive outcome of assisted fertilization.

## 5. Materials and Methods

### 5.1. Samples

Seven ejaculated semen samples from donors (aged 32 to 40 years) were analyzed in this in vitro study. Samples were obtained after 3–5 days of sexual abstinence. All donors adhered to the study protocol and were informed of their privacy and that their samples would not be used for fertilization. Signed informed consent was provided by all donors. Approval of the Ethics Committee of Azienda Ospedaliera Universitaria Senese (CEAOUS) was not required. 

Sperm analysis was carried out following WHO guidelines [17]. 

### 5.2. Chemical Structures of 4(RS)-4-F_4t_-NeuroPs, 10(S)-10-F_4t_-NeuroP and 10(R)-10-F_4t_-NeuroP

As 4- and 10-F_4t_-NeuroPs are not commercially available, they were synthesized in-house for the study. The chemical structures of 4- and 10-F_4t_-NeuroPs [36,37] are shown in Figure 3.

### 5.3. Swim-Up Technique on Human Semen Samples and Incubation in Presence of F_4_-NeuroPs

A swim-up technique was used to obtain the motile sperm fraction: 0.5 mL of each semen sample was placed in a sterile conical centrifuge tube and gently layered with 0.5 mL of Sperm Washing Medium (Irvine Scientific, Santa Ana, CA, USA). The tubes, inclined at a 45° angle, were incubated for 45 min at 37°C under 5% CO_2_. Then, 0.5 mL of the uppermost medium that contained the motile sperm fraction was collected and used for the experiments. Approximately 10 × 10^6^ swim-up selected sperm/mL were divided into aliquots and incubated at 37 °C under 5% CO_2_ as follow:-7 ng F_4_-NeuroPs for 2 h-7 ng F_4_-NeuroPs for 4 h

In each incubation experiment, 7 ng F_4_-NeuroPs was considered to be the final quantity of F_4_-NeuroPs. This F_4_-NeuroP amount was used according to previous work in which this amount was tested [14]. To add 7 ng F_4_-NeuroPs in each set of incubation systems, a 3 ng/mL stock solution of F_4_-NeuroPs (1:1 mixture of 4-F_4t_-NeuroP and 10-F_4t_-NeuroP) was prepared in ethanol and a volume was added to each sample to obtain the final amount of 7 ng. For each sample, the final percentage of ethanol in the incubation mixture was no more than 0.07%. Control samples were incubated in the absence of F_4_-NeuroPs, but in the presence of a comparable percentage (as referred to the sample volume) of ethanol.

Swim-up selected sperm treated with the same conditions and times, but without F_4_-NeuroPs, were used as controls.

After incubation, sperm motility, vitality, DNA susceptibility to damage, MMP, acrosome integrity, and the immunolocalization of ryanodine receptor were evaluated. 

Sperm motility was evaluated using a computer-assisted sperm analyzer (CASA; ISAS model, Valencia, Spain). Two drops and six microscopic fields were set up for a minimum of 300 sperm tracks. All semen samples were recorded at 100 Hz frames for 1 s; thus, 12–200 successive images were recorded.

Motility percentage, curvilinear velocity, straight-line velocity, average path velocity, linearity, amplitude of lateral head displacement, and beat cross frequency were also assessed.

### 5.4. Acridine Orange Assay

The acridine orange (AO) assay was used to evaluate DNA susceptibility to denaturation following the protocol described by Tejada et al. [38]. A stock solution of 1% AO (3, 6-bis [dimethylamino] acridine, hemi [zinc chloride] salt; BDH Chemicals Ltd., Poole, England) was prepared and stored in the dark at 4 °C until use. The working solution was prepared by mixing 1 mL of AO stock solution with 4 mL of 0.1 M citric acid (C_6_H_8_O_7_) and 0.250 mL of 0.3 M disodium hydrogen phosphate heptahydrate (Na_2_HPO_4_ 7H_2_O). First, samples were centrifuged at 400× *g* for 10 min, then the pellet was reserved and resuspended in phosphate-buffered saline (PBS). A drop of each sample was smeared onto a glass slide and subsequently fixed overnight in Carnoy’s fixative (methanol:acetic acid 3:1). Slides were washed in PBS, stained for 5 min, washed in distilled water, and finally observed under a Leitz Aristoplan fluorescence microscope (Leica, Wetzlar, Germany) equipped with a 490 nm excitation light and 530 nm barrier filter. In each sample, more than two hundred spermatozoa were evaluated. The presence of a green fluorescence label indicated a sperm with double-stranded DNA (dsDNA), whereas sperm was evaluated to have denatured DNA (ssDNA) if the fluorescence label was red.

### 5.5. JC-1 Dye

JC-1 dye (Molecular Probes, Eugene, OR, USA) enables evaluation of the membrane mitochondrial potential. After treatment, swim-up selected sperm were centrifuged at 400× *g* for 10 min, then the obtained pellet was reserved and dark-incubated with a solution of JC-1 dye (1 μg/mL in PBS) for 20 min at 37 °C. Cells were centrifuged, resuspended in PBS, and examined under a Leitz Aristoplan fluorescence microscope (Leica, Wetzlar, Germany) equipped with a 490 nm excitation light and 530 nm barrier filter. Sperm with healthy mitochondria displayed red fluorescence in the presence of JC-1 complexes known as J-aggregates. In sperm with unhealthy mitochondria, JC-1 was present in monomeric form and emitted a diffuse green fluorescence. The fluorescence signal was evaluated as red spots and strong red signals [39].

### 5.6. Pisum Sativum Agglutinin

Pisum sativum agglutinin (PSA) is a lectin with specificity towards acrosome-associated glycoconjugates and was used to observe acrosome status. This was possible because of the presence of a TRITC fluorochrome linked to PSA. First, we washed the cells in PBS twice for 5 min each time. After this, samples were incubated with 1 μg/mL PSA (in PBS) for 30 min, washed with PBS for 15 min, and nuclei were stained with 4,6-diamidino-2-phenylindole (DAPI) solution (Vysis, Downers Grove, IL, USA) diluted 1:20,000 for 10 min. The samples were washed twice more with PBS for 15 min each time, and 1,4-diazabicyclo [2.2.2]octane (DABCO) was added an antifading agent. Slides were covered with a coverslip, sealed, and observed under a Leitz Aristoplan fluorescence microscope (Leica, Wetzlar, Germany). Intact acrosomes displayed intense red fluorescence, while sperm with reacted acrosomes did not display red fluorescence or were only labeled at the equatorial segment.

### 5.7. Immunofluorescence of Ryanodine Receptor

Swim-up selected spermatozoa were washed with PBS, smeared on glass slides, air dried, and fixed in 4% paraformaldehyde for 15 min. After treatment with blocking solution (1% bovine serum albumin [BSA], 5% normal goat serum [NGS]) for 20 min, the slides were incubated overnight at 4 °C with a primary anti-ryanodine receptor polyclonal antibody (Invitrogen, Thermo Fisher Scientific, Carlsbad, CA, USA) diluted 1:500. The next day, the samples were washed 3 times in PBS with 0.1% Tween 20 and treated with a secondary anti-rabbit antibody raised in goat Alexa Fluor^®^ 488 conjugate (Invitrogen, Thermo Fisher Scientific, Carlsbad, CA, USA) diluted 1:500 for 1 h at room temperature. Nuclei were stained with DAPI solution (Vysis, Downers Grove, IL, USA) for 10 min, and the slides were washed in PBS and mounted with DABCO (Sigma-Aldrich, Milan, Italy) in order to observe fluorescence. To reveal non-specific binding of the secondary antibody, a negative control was created by omitting the primary antibody. The slides were observed under a Leica DMI 6000 fluorescence microscope (Leica Microsystems, Wetzlar, Germany) and images were acquired using the Leica AF 6500 Integrated System for Imaging and Analysis. For each sample, approximately two hundred sperm were evaluated and the presence of labeling was recorded.

### 5.8. Statistical Analysis 

Statistical analysis was performed using IBM SPSS version 23.0 for Windows software package (IBM Corp., Armonk, NY, USA). The appropriate analysis in this study, where the variables size was less than 30, was to use non-parametric tests. To determine the homogeneity of the variances, Levene’s test was applied. The Mann-Whitney U test was used to compare differences between two independent groups. The Kruskal-Wallis test (one-way ANOVA on ranks) was performed for multiple comparisons of more than two independent groups, followed by post-hoc comparisons using the Dunnett test or Tukey test. The variables were reported as median (interquartile range [IQR]). A value of *p* < 0.05 was considered statistically significant.

## Figures and Tables

**Figure 1 ijms-24-00935-f001:**
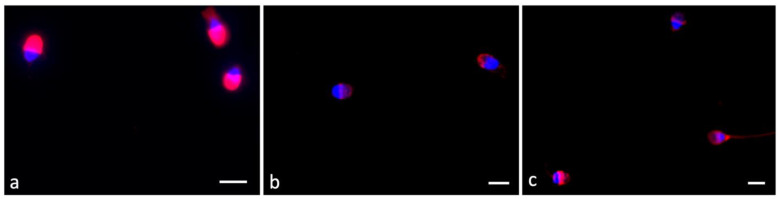
Evaluation of the acrosome status with TRITC–pisum sativum agglutinin (TRITC–PSA). Intact acrosome (**a**), reacted or partially reacted acrosome (**b**,**c**) with no fluorescence or only fluorescence of the equatorial segment and loss of acrosomal membranes. Bars: 6 μM.

**Figure 2 ijms-24-00935-f002:**
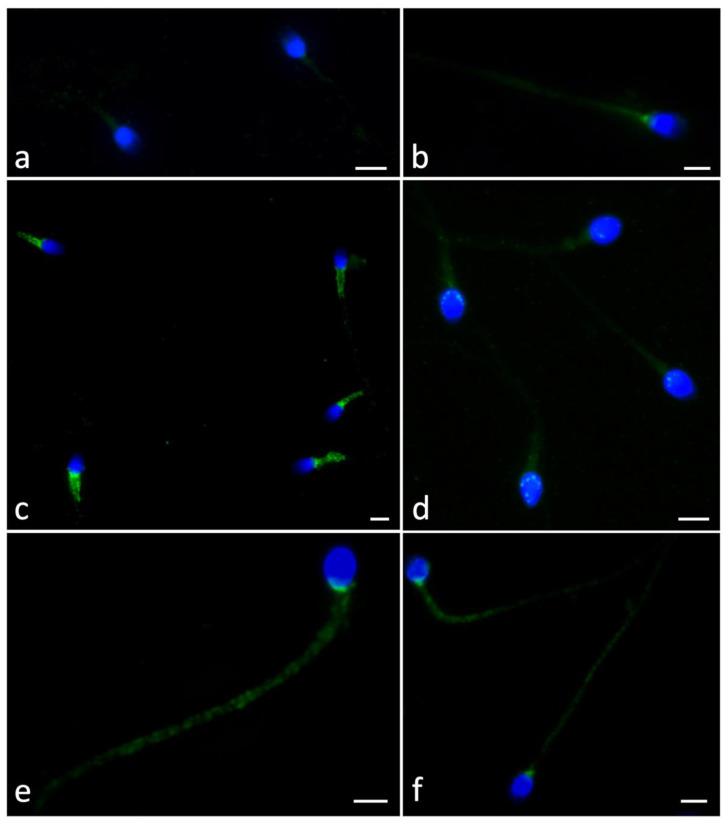
UV micrographs of swim-up selected sperm treated with anti-ryanodine antibody. The percentage of localization differed in control samples or those incubated with F_4_-NeuroPs for 2 or 4 h. In the controls, a high percentage of sperm showed diffuse and fair labeling (**a**) or a spot localized in the centriolar region (**b**). Sperm incubated for 2 h with F_4_-NeuroPs showed strong signals in the midpiece and postacrosomal sheath (**c**,**d**). After 4 h F_4_-NeuroP incubation, the labeling appeared dotted along the total tail (**e**,**f**). Nuclei (blue) were stained with DAPI. Figure 2 without DAPI was included in Appendix A. Bars: 6 μM.

**Figure 3 ijms-24-00935-f003:**
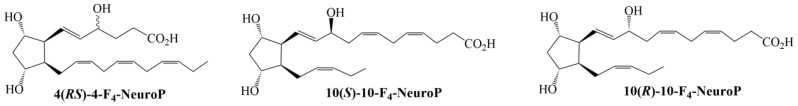
Representation of chemical structures of 4- and 10-F_4t_-NeuroP isomers.

**Table 1 ijms-24-00935-t001:** Median (IQR) percentage of total progressive sperm motility evaluated by computer-assisted sperm analyzer (CASA) after incubation with 7 ng F_4_-NeuroPs and without F_4_-NeuroPs (control) for 2 and 4 h. Statistics are also reported; *p* < 0.05 is considered significant.

	Control (2 h)	F_4_-NeuroPs (2 h)	Control (4 h)	F_4_-NeuroPs (4 h)
Sperm progressive motility%	44 (42–46)	56 (52–64)	42 (40–43)	44 (42–48)
Statistics	*p* = 0.02		*p* = 0.119	

**Table 2 ijms-24-00935-t002:** Median (IQR) percentage of sperm with JC-1 labeling (high MMP potential), showing as a red spot or strong red signal after incubation with 7 ng F_4_-NeuroPs or without F_4_-NeuroPs (control) for 2 and 4 h. *p* < 0.05 is considered significant.

	Control 2 h Red Spot	F_4_-NeuroPs 2 h Red Spot	Control 2 h Red Spot Strong Signal	F_4_-NeuroPs 2 h Red Spot Strong Signal	Control 4 h Red Spot	F_4_-NeuroPs 4 h Red Spot	Control 4 h Red Spot Strong Signal	F_4_-NeuroPs 4 h Red Spot Strong Signal
J-C1-labeled sperm%	68 (65–74)	78 (74–79)	26 (24–28)	50 (48–52)	44 (42–46)	42 (42–43)	18 (16–18)	20 (18–21)
Statistics	*p* = 0.012		*p* = 0.002		*p* = 0.173 (ns)		*p* = 0.042	

**Table 3 ijms-24-00935-t003:** Median (IQR) percentage of swim-up selected sperm with acrosome intact, partially reacted, or totally reacted after incubation with 7 ng F_4_-NeuroPs or without F_4_-NeuroPs (control) for 2 and 4 h. Statistics are also reported; *p* < 0.05 is considered significant.

	Control 2 h (C2)	F_4_-NeuroPs 2 h (F_4_-2)	Control 4 h (C4)	F_4_-NeuroPs 4 h (F_4_-4)	Statistics
sperm with intact acrosome %	82 (81–83.50)	56 (55–58)	58 (57–59)	15 (14.50–15.50)	C2 vs. F_4_-2, *p* = 0.002 C4 vs. F_4_-4, *p* = 0.002 C2 vs. C4, *p* = 0.001 F_4_-2 vs. F_4_-4, *p* = 0.001
sperm with partially reacted acrosome %	12 (11–14)	18 (17–19)	22 (21–22)	44 (43–44.50)	C2 vs. F_4_-2, *p* = 0.263 C4 vs. F_4_-4, *p* = 0.001 C2 vs. C4, *p* = 0.091 F_4_-2 vs. F_4_-4, *p* = 0.001
sperm with totally reacted acrosome %	5 (4.50–5.50)	24 (23–24)	20 (20–21)	40 (40–42)	C2 vs. F_4_-2, *p* = 0.001 C4 vs. F_4_-4, *p* = 0.003 F_4_-2 vs. F_4_-4, *p* = 0.005

**Table 4 ijms-24-00935-t004:** Median (IQR) percentage of swim-up selected sperm treated with anti-ryanodine antibody after incubation with 7 ng F_4_-NeuroPs or without F_4_-NeuroPs (control) for 2 and 4 h. Statistics are also reported; *p* < 0.05 is considered significant.

	Control (2 h)	F_4_-NeuroPs (2 h)	Control (4 h)	F_4_-NeuroPs (4 h)	Statistics
sperm with diffuse fair signal%	62 (60.50–63.50)	10 (10–13)	23 (22–24)	10 (8.50–13)	C2 vs. F_4_-2, *p* = 0.001 C4 vs. F_4_-4, *p* = 0.001 C2 vs. C4, *p* = 0.001 F_4_-2 vs.F_4_-4, *p* = 0.981
sperm with absence of signal%	23 (20.25–25.75)	10 (10–13)	12 (10.50–13.50)	14 (10.75–15)	C2 vs. F_4_-2, *p* = 0.001 C4 vs. F_4_-4, *p* = 0.865 C2 vs. C4, *p* = 0.001 F_4_-2 vs.F_4_-4, *p* = 0.527
sperm with mitochondrial signal %	6 (4.25–9.25)	70 (68–73.50)	38 (36.50–39.50)	25 (25–25.75)	C2 vs. F_4_-2, *p* = 0.001 C4 vs. F_4_-4, *p* = 0.001 C2 vs. C4, *p* = 0.001 F_4_-2 vs.F_4_-4, *p* = 0.001
sperm with dotted signal in tail	8 (8–9.50)	9 (6.50–11.50)	27 (26–28)	53.50 (50.50–56.50)	C2 vs. F_4_-2, *p* = 0.999 C4 vs. F_4_-4, *p* = 0.001 C2 vs. C4, *p* = 0.001 F_4_-2 vs.F_4_-4, *p* = 0.001

## Data Availability

The data that support the findings of this study are available from the corresponding author upon reasonable request.

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
