# Peer review of "F4-Neuroprostane Effects on Human Sperm"

_ijms, 2023, doi:10.3390/ijms24020935_

Round 1

Reviewer 1 Report

The aim of this study was to verify the F4 Neuroprostane effect on human sperm. I find this manuscript as a good example of an original paper describing the effect of F4 Neuroprostane acting on human sperm quality. From my point of view the draft is complete and intelligible. A sufficient number of samples as well as up-to-date analytical and statistical methods were used in the study. Moreover, all chapters of mentioned paper have been written in a lucid and comprehensive way. Presented data does not raise any objections and conclusion clearly emerges from the results obtained by the authors.

 Nevertheless, “Materials and Methods” chapter should be slightly revised. The description of the method that was used to determine the progressive movement of sperm should be provided. Did the author employ the objective method as CASA system or not?

Reviewer 2 Report

The work is well planned and the experiments have been correctly designed to answer the suggested hypothesis. However, I consider that some issues should be clarified to ensure the understanding and reliability of the work.

 Other comments are on a personal level about doubts that may arise as a result of reading the pre-proof.

Comments:

1- In the introduction it is pointed out that Fertile men highlighted lower levels of seminal F4-NeuroPs compared to those detected in infertile men due to different pathological reproductive conditions and a positive correlation was detected with sperm necrosis, however, it seems that addiction in vitro of seminal F4-NeuroPs could help increase the fertilizing capacity of sperm. What could cause this difference in vivo/in vitro? Could you explain more about this in the article or in this comment?

2- Some references to similar articles regarding the methodology should be added to explain some points. For example, there are authors who point out that spermatozoa labeling with PNA is indicative of the integrity of the acrosome membrane and might reflect both damaged and reacted acrosome. This should be discussed when the acrosomal reaction is mentioned as an indicator of capacitation.

3- Following the above, it should be mentioned (if done in this way) that in the same sample after the treatments, the viability of the spermatozoa has been carried out and correlated with those that show an acrosomal reaction, to clarify this point.

About results:

4- Because images of staining with JC1 have not been included

5- In figure 2, you could include control images where the same spermatozoa have been stained with DAPI to visualize the nuclei and it would also be good to include bright field images as in the related work (doi:10.4103/1008682X.185848) that follows a similar methodology. This reference must be considered. 

About methods:

6- How has motility been measured and what parameters have been evaluated (C.A.S.A, counting chamber?)

7- Has the viability been carried out in the same sample of the treatments to be correlated to the PNA markings?

Other questions and discussion:

8- Do you think that the decrease in progressive motility at four hours is related to the change in motility parameters (LIN, STR, WOB...) due to the acrosome reaction?

9- In the discussion, you add that MMP is positively correlated with sperm motility and negatively with lipid peroxidation in bovine. It should be added (add references) that in humans there is no consensus on whether MMP is associated with sperm quality or the difference in MMP between capacitated and non-capacitated spermatozoa.

10- -          Line 189- Add “) at the end of the sentence
